# On the Implementation of Simultaneous Multi-Frequency Excitations and Measurements for Electrical Impedance Tomography [note 1]

**DOI:** 10.3390/s19173679

**Published:** 2019-08-24

**Authors:** Mathieu Darnajou, Antoine Dupré, Chunhui Dang, Guillaume Ricciardi, Salah Bourennane, Cédric Bellis

**Affiliations:** 1CEA Cadarache, 13115 Saint-Paul-lez-Durance, France; 2École Centrale de Marseille, 38 Rue Frédéric Joliot Curie, 13013 Marseille, France; 3Department of Paediatrics, University of Geneva, 1205 Geneva, Switzerland; 4Aix Marseille Univ, CNRS, Centrale Marseille, LMA UMR 7031, Marseille, France

**Keywords:** FPGA, high-speed EIT, frequency division multiplexing, ONE-SHOT, EIDORS

## Abstract

The investigation of quickly-evolving flow patterns in high-pressure and high-temperature flow rigs requires the use of a high-speed and non-intrusive imaging technique. Electrical Impedance Tomography (EIT) allows reconstructing the admittivity distribution characterizing a flow from the knowledge of currents and voltages on its periphery. The need for images at high frame rates leads to the strategy of simultaneous multi-frequency voltage excitations and simultaneous current measurements, which are discriminated using fast Fourier transforms. The present study introduces the theory for a 16-electrode simultaneous EIT system, which is then built based on a field programmable gate array data acquisition system. An analysis of the propagation of uncertainties through the measurement process is investigated, and experimental results with fifteen simultaneous signals are presented. It is shown that the signals are successfully retrieved experimentally at a rate of 1953 frames per second. The associated signal-to-noise ratio varies from 59.6–69.1 dB, depending on the generated frequency. These preliminary results confirm the relevance and the feasibility of simultaneous multi-frequency excitations and measurements in EIT as a means to significantly increase the imaging rate.

## 1. Introduction

In the context of high pressure and high temperature flow rigs, flow patterns are investigated. Tomography is a suitable technique for its applicability and non-intrusiveness. This imaging technique is used to reconstruct the volume of an object from measurements of penetrating waves or particles on its external boundary. Tomography is based on measurements of the boundary conditions of a system, where both the signal emitter and receiver have to be well known in space. Various tomography techniques exist using electromagnetic, acoustic waves, or electron and muon particles. Among the use of tomography in applications, the most well established is medical imaging using X-ray-based CT scans or electromagnetic waves in magnetic resonance imaging. Nevertheless, these techniques are not robust in this context, complex to implement, and expensive.

Electrical Impedance Tomography (EIT) [1,2,3,4] is another tomographic method that consists of imposing (resp. measuring) an electrical current passing through a set of electrodes at the surface of a body, while measuring (resp. imposing) the electrical potential over another set of electrodes. This technique is preferred for its applicability at high pressure and temperature, robustness, and relatively low cost. The standard approach of EIT is based on the concept of Time-Division Multiplexing (TDM), where the excitation signal is routed to a single pair of electrodes at any given time. The sequential selection of these pairs of electrodes with multiplexers or electronic switching [5] creates an excitation strategy and builds an EIT data frame, which includes the measurement data for all excitation pairs. The EIT data frame defines the Neumann and Dirichlet boundary conditions to be used in the associated inverse problem of reconstructing the electrical conductivity field inside the body.

EIT necessitates potentially a high data-frame acquisition rate, for instance in flow study involving rapidly-evolving flow regimes [6,7]. The challenge is that increasing the number of frames per seconds (fps) reduces the measurement time, while a large number of measurements is required to limit the ill-posedness of the inverse problem. In addition, the sequential excitation of the electrodes in contact with ionic solutions results in significant difficulties caused by transient voltages due to the presence of capacitive components in the electrode–electrolyte interface [8]. Therefore, the TDM sequential measurement procedure requires dead time, complex signal processing techniques, or hardware improvement to limit the effects of contact impedance [9,10].

Several high-speed EIT systems of hundreds of fps have been proposed in the literature. In [11], a multichannel architecture system was presented, containing a control module that manages and synchronizes 64 channels capable of generating and measuring voltages and currents. The system was able to collect 182 frames per seconds when acquiring 15 spatial patterns with a Signal-to-Noise Ratio (SNR) from 65.5–96 dB when averaging the signal from 32 samples. Another EIT system proposed in [12] allows capturing over 100 frames per seconds with an SNR greater than 90 dB when averaging the signal. Concerning Electrical Capacitance Tomography (ECT), another soft field imaging technique, several systems were also reported in [13,14,15] with high frame rate. Furthermore, an EIT system was developed in our group [16] to reach the Data Acquisition (DAQ) rate of 833 fps when acquiring 120 spatial patterns with TDM. Nevertheless, in any case, high speed is reached at the price of a small number of electrodes, a partial scanning strategy, or short measurement times.

Equivalent to TDM, in the early stages of X-ray tomography, a single source/detector pair was rotated around a body to be imaged, resulting in a low acquisition rate. The improvement of X-ray tomography systems led to multiple emitter/receiver pairs that do not interact with each other as X-rays pass straight through the body. The sources and receivers are activated simultaneously, improving significantly the acquisition rate. By analogy with the evolution of X-ray tomography, a solution for increasing the speed of EIT is to perform simultaneous measurements. However, the sensitivity regions within the body depend on the conductivity of the material, which is a priori unknown since it depends on the conductivity distribution inside the body, which is unknown. Therefore, simultaneous electrical emissions and measurements through surface electrodes result in a superposition of the signals in the measurement channels.

The ONe Excitation for Simultaneous High-speed Operation Tomography (ONE-SHOT) is an innovative method developed by our group [17] to give a solution for the problem of the superposition of signals for simultaneous excitation in EIT systems. Based on simultaneous multi-frequency stimulations, the overlapping of signals is discriminated using Frequency-Division Multiplexing (FDM) techniques. FDM, used in telecommunications, consists of dividing the total bandwidth available into a series of non-overlapping frequency bands, each of which is used to carry a separate signal. FDM is used in the context of EIT [18,19], and simultaneous excitations at different frequencies can be discriminated by demodulation, resulting in simultaneous excitations and measurements.

The ONE-SHOT approach brings several novelties in the field of FDM EIT. Firstly, in the excitation strategy of the previous systems, half of the electrodes are tagged with an excitation frequency, and the other half is used for measurement: the electrodes are either used for excitation or measurement. The ONE-SHOT method provides current measurement in the excitation circuit, and each electrode is simultaneously used for both excitation and measurement. Secondly, in the previous systems, the current excitation electrodes are tagged with a single frequency. On the other hand, ONE-SHOT introduces in this article an experiment where 15 excitation signals at 15 frequencies are imposed on a single electrode. It aims at demonstrating the applicability of FDM in this situation. Thirdly, a central argument in using FDM is the absence of transients at the electrode–electrolyte, interface resulting in a high data frame rate and low noise. The association of continuous excitations with a point-by-point synchronous Fourier transform is discussed to optimize the data acquisition speed. In addition, an innovative hardware method leads to the implementation of every excitation and measurement, including real-time fast Fourier transform on a single FPGA chip.

The present article proposes a practical implementation of the ONE-SHOT in a physical experiment to prove the feasibility of the method. The method was introduced in [17] for four electrodes; however, in practice, EIT systems usually contain eight electrodes or more for better performances. After an overview of the mathematical aspect of EIT in Section 2, a first step is to determine the adequate number of electrodes for satisfying performances and to adapt the theory of ONE-SHOT consequently (Section 3). Secondly, the high frame rate generates a large amount of data to process. In the past few years the development of electronics made it possible to handle systems allowing large measurement data to transfer [20]. The estimation of the optimal data transfer rate and online computation speed is needed to choose the adequate hardware components, an issue that is discussed in Section 4. Thirdly, the error propagation through the FDM is estimated in this context based on two quantitative numerical simulations (Section 5). Finally, the article presents preliminary results of the demodulation of 15 simultaneous excitation frequencies in Section 6.

## 2. Electrical Impedance Tomography

EIT is a type of non-intrusive technique widely used in medical, geo-physical, and industrial imaging. From the information of electrical current and potential on the boundary of an object, the Calderón’s inverse problem aims at determining the conductivity distribution inside the object. This section introduces the mathematical aspects of EIT as generally reported in the literature.

The system $\Omega \subset R^{3}$ in which the image reconstruction is considered is cylindrical (Figure 1). EIT is a technique to solve the inverse problem of determining the conductivity function γ(x,ω)=σ(x)+iωϵ(x) within the interior of Ω [2,3,4]. Here, σ(x) is the isotropic electrical conductivity field, ϵ(x) the permittivity, and ω the angular frequency. EIT operates at ω≪1 GHz, so the imaginary part of γ can be neglected γ(x,ω)→σ(x). The closure of Ω is denoted as $\bar{\Omega }$, and its boundary is ∂Ω.

In EIT, the conductivity function σ(x) is unknown and has to be determined from simultaneous excitations and measurements of the voltage u(x) and the current j(x)=σ(x)∇u, on the boundary. The Maxwell equations define j(x) to be divergence free. Thus, Ohm’s law gives the partial differential equation:(1)∇·j(x)=0forx∈Ω.
The EIT detector contains a set of $n_{e}$ electrodes. On each electrode $E_{n}\subset \partial \Omega$ for $0\leq n\leq n_{e}$ (Figure 1b), the potential $V_{n}$ and the current $I_{n}$ are either given or measured, defining the boundary conditions. Equation (Equation 1) is taken with either the *Dirichlet boundary conditions*: (2)$$u\left(x\right)\;=\;\left\{\begin{array}{cc} V_{n}\left(x\right) & \mathrm{for}x\in E_{n},\\ 0 & \mathrm{for}x\in \partial \Omega \;\mathrm{and}\;x\notin E_{n}, \end{array}\right.$$
or the *Neumann boundary conditions*: (3)$$j\cdot n\left(x\right)\;=\;\left\{\begin{array}{cc} I_{n}\left(x\right) & \mathrm{for}x\in E_{n},\\ 0 & \mathrm{for}x\in \partial \Omega \;\mathrm{and}\;x\notin E_{n}, \end{array}\right.$$
where n(x) is the unitary vector on the surface pointing outward at x∈∂Ω. The Neumann boundary conditions have the additional requirement that:(4)$$\sum_{n=1}^{n_{e}}\;I_{n}\left(x\right)\;=\;0,$$
as required by Equation (Equation 1) for well-posedness. In addition, in the case of imposed potentials, the Dirichlet boundary conditions are chosen with the average-free boundary voltage requirement:(5)$$\sum_{n=1}^{n_{e}}\;V_{n}\left(x\right)\;=\;0,$$
in order to define the ground value.

The so-called Dirichlet-to-Neumann (DtN) map $\Lambda_{\sigma }:u{{|}_{\partial \Omega }⟼\sigma \left(x\right)\nabla u\cdot n|}_{\partial \Omega }$ is the operator that links the imposed potential (i.e., Equation (Equation 2)) to the current on the boundary. Equivalently, the Neumann-to-Dirichlet (NdD) map $R_{\sigma }:\sigma \nabla u\cdot n{{|}_{\partial \Omega }⟼u|}_{\partial \Omega }$ yields the voltage distribution from any given current-density distribution on the boundary (i.e., Equation (Equation 3)). The DtN or NtD map contains the information to determine σ(x) from the set of $V_{n}\left(x\right)$ (or $I_{n}\left(x\right))$. The aim of EIT is to reconstruct the conductivity distribution σ(x) while **imposing** either the *Dirichlet* or the *Neumann* boundary conditions while **measuring** respectively the *currents* or the *potentials* over the electrodes [21,22]. Finally, the measurements relative to different excitation patterns constitute the data frame, which is used to solve the associated inverse problem to reconstruct σ(x).

## 3. Specifications of the Proposed EIT System

The ONE-SHOT method was introduced in [17], which contains motivating results that predict the feasibility of the demodulation of simultaneous excitations with respect to their frequencies. Nevertheless, the proof-of-principle experiment is based on a four-electrode EIT system. The number of independent pairs of excitation for such systems is six, implying six simultaneous excitations at six different frequencies. In practical EIT, the number of electrodes is an important parameter since it allows more measurements and a better conditioning of the inverse problem. Usually, EIT systems contain more than eight electrodes.

The need for more simultaneous measurements is the motivation to evolve the ONE-SHOT method with the increased number of electrodes and, consequently, the number of excitation frequencies. This section discusses the choice of developing the ONE-SHOT excitation strategy for 16 electrodes in Section 3.1. Secondly, the ONE-SHOT excitation strategy is generalized to any number of electrodes in Section 3.2 and then adapted for 16 electrodes in Section 3.3. This process increases significantly the number of independent pairs and the complexity of the corresponding excitation patterns. Finally, choosing the frequencies is a challenging task. A choice is proposed in Section 3.4 to ensure discriminability, a high rate, and adaptivity to high-speed hardware systems.

### 3.1. Adequate Number of Electrodes

A large number of measurements is required to well condition the inverse problem for an accurate and stable solution. Consider an EIT device containing a set of $n_{e}$ electrodes. The maximum number of measurements is:(6)$$M\;=\;N\times n_{e},$$
where *N* is the total number of independent excitation pairs, i.e.:(7)$$N\;=\;\frac{n_{e}\;\left(n_{e}-1\right)}{2}.$$

As $M∼n_{e}^{3}$, the dataset size increases rapidly as the number of electrodes increases. One can expect that larger values of the parameter $n_{e}$ lead to images of better quality.

To evaluate this, the open access code EIDORS (Electrical Impedance Tomography and Diffuse Optical Tomography Reconstruction Software) [23] was used. The code provides software algorithms for forward and inverse modeling for EIT and diffusion-based optical tomography, in medical and industrial settings. Synthetic data of several flow patterns with EIT detectors containing $n_{e}=$ 8, 16, and 32 electrodes are computed. The data are associated with an image based on the linear back projection reconstruction algorithm. This non-iterative method projects the set of voltage variations $\left[\delta V_{n}\right]$ between homogenous and an inhomogeneous conductivity data onto the maps of conductivity change δσ(x) with a set of sensitivity coefficients *S* calculated with a linearized version of the inverse problem [24].

In the simulations of Figure 2, the domain is filled with homogenous water of conductivity 0.5 S·m^−1^ with the addition of steam bubbles of conductivity $1\times 10^{-5}$ S·m^−1^. Three patterns are represented in the figure. The number of measurements is M=224 for $n_{e}\;=\;8$, M=1920 for $n_{e}\;=\;16$, and M=15,872 for $n_{e}\;=\;32$. For every pattern, increasing $n_{e}$ improves the image quality, as expected. However, the situation $n_{e}\;=\;32$ does not give much more accuracy than $n_{e}\;=\;16$, even if the number of measurements is multiplied by more than eight. This result is a known matter in the field of EIT.

Any system with a high frame rate brings an important amount of data. The balance between high image accuracy and low data size led us to develop an EIT system with a specific number of 16 electrodes. According to Equation (Equation 7), the full set of excitations for an EIT system of 16 electrodes is N=120.

### 3.2. General Simultaneous Excitation Pattern for $n_{e}$ Electrodes

ONE-SHOT has to be adapted for a larger number of electrodes. As in Equation (Equation 7), the total number of independent measurements *N* defines the number of frequencies that have to be generated simultaneously to maximize the number of measurements for a given system. The excitation is a set of voltages imposed on the electrodes and is generated from a basis of *N* sines. Moreover, as discussed in Section 2 the excitation pattern ensures that the sum of boundary voltages is zero at any time.

One defines the signal $\Psi_{i}$ as:(8)$$\Psi_{i}\left(t\right)=A\sin\left(2\pi f_{i}t\right).$$
Then, given an arbitrary number $n_{e}$ of electrodes, one defines the excitation voltage $V_{n}\left(t\right)$ at an arbitrary electrode $n\in \{1,\ldots ,n_{e}\}$ using the following recurrence relation:(9)$$V_{n}\left(t\right)=-\sum_{j=1}^{n-1}{\left[V_{j}\left(t\right)\right]}_{n-1}+\sum_{\ell =\ell_{n}^{\min}}^{\ell_{n}^{\max}}\Psi_{\ell }\left(t\right)$$
where:(10)$$\ell_{n}^{\min}=\left(n-1\right)n_{e}-\frac{n(n-1)}{2}+1\;\mathrm{and}\;\ell_{n}^{\max}=n\;n_{e}-\frac{(n+1)n}{2}$$
and with ${\left[V_{j}\left(t\right)\right]}_{n-1}$ designating the ${\left(n-1\right)}^{\mathrm{th}}$ element of the identity defining the voltage $V_{j}\left(t\right)$, under the convention that the terms $\Psi_{i}$ are always ordered with increasing index *i* in such an identity. Moreover, in (Equation 9), we adopt the convention that a sum is identically zero if the value of the starting index is larger than that of the ending one.

### 3.3. Simultaneous Excitation Pattern for 16 Electrodes

We are now interested in applying Equation (Equation 9) in the situation of $n_{e}=16$ electrodes. In this situation, N=120 and the set of excitation voltages $\left[V_{n}\left(t\right)\right]$ is: (11)$$\left\{\begin{array}{ccccccccc} \begin{array}{c} V_{1}\left(t\right) \end{array} & \;=\; & +\Psi_{1} & +\Psi_{2} & +\Psi_{3} & + & \;\ldots & +\Psi_{14} & +\Psi_{15}\\ V_{2}\left(t\right) & \;=\; & -\Psi_{1} & +\Psi_{16} & +\Psi_{17} & + & \;\ldots & +\Psi_{28} & +\Psi_{29}\\ V_{3}\left(t\right) & \;=\; & -\Psi_{2} & -\Psi_{16} & +\Psi_{30} & + & \;\ldots & +\Psi_{41} & +\Psi_{42}\\ V_{4}\left(t\right) & \;=\; & -\Psi_{3} & -\Psi_{17} & -\Psi_{31} & + & \;\ldots & +\Psi_{53} & +\Psi_{54}\\ & \vdots \\ V_{15}\left(t\right) & \;=\; & -\Psi_{14} & -\Psi_{28} & -\Psi_{41} & - & \;\ldots & -\Psi_{118} & +\Psi_{120}\\ V_{16}\left(t\right) & \;=\; & -\Psi_{15} & -\Psi_{29} & -\Psi_{42} & - & \;\ldots & -\Psi_{119} & -\Psi_{120} \end{array}\right.$$
where the frequency $f_{120}$ is the $N^{\mathrm{th}}$ frequency for $n_{e}\;=\;16$. Finally, the verification of Equation (Equation 5) from the excitation pattern of Equation (Equation 11) is straightforward.

### 3.4. Determination of the Excitation Frequencies Based on the Measurement Time Window

The main advantage in the use of a continuous multi-frequency excitation method is the absence of transients between successive projections, resulting in a diminution of the measurement error from the absence of the residual voltage in the electrode–electrolyte contact impedance. Concerning ONE-SHOT, the values of the frequencies remain as free parameters. In this section, we suggest suitable values based on the performances of the DAQ system.

The first observation is that in order to ensure continuous excitations and measurements in parallel, the frequencies of the excitation signals have to be harmonics of a fundamental frequency $f_{0}$. A wise choice is to define $f_{0}$ as the frequency of the Discrete Fourier Transform (DFT) of a *P*-point measurement sequence $\left\{V_{n}\left(p\right)\right\}$ and *p* the discrete time. The discretisation of the time is due to the sampling rate of the DAQ system at the frequency $f_{DAQ}$, with the discretisation time interval $\Delta p=1/f_{DAQ}$, implying:(12)$$f_{0}\;=\;\frac{f_{DAQ}}{P}.$$

The DFT of the real-valued sequence $\left\{V_{n}\left(p\right)\right\}$ in the $1\leq n\leq n_{e}$ measurement channel is:(13)$${\hat{V}}_{n}\left(k\right)\;=\;\frac{1}{P}\sum_{p=0}^{P-1}V_{n}\left(p\right)e^{-ik\beta_{p}},\;k=0,\ldots ,P-1,$$
with $\beta_{p}\;=\;2\pi p/P$, and a synchronous sampling is assumed. The real and imaginary parts of ${\hat{V}}_{n}\left(k\right)$:(14)$$R_{n}\left(k\right)\;=\;\frac{1}{P}\sum_{p=0}^{P-1}V_{n}\left(p\right)\cos\left(k\beta_{p}\right)$$
and:(15)$$I_{n}\left(k\right)\;=\;-\frac{1}{P}\sum_{p=0}^{P-1}V_{n}\left(p\right)\sin\left(k\beta_{p}\right)$$
define the module $M_{n}\left(k\right)$ and the phase $\phi_{n}\left(k\right)$ of each frequency domain sample ${\hat{V}}_{n}\left(k\right)$:(16)$$M_{n}\left(k\right)\;=\;\sqrt{R_{n}^{2}\left(k\right)+I_{n}^{2}\left(k\right)}$$
and:(17)$$\phi_{n}\left(k\right)\;=\;\arctan\left(\frac{I_{n}\left(k\right)}{R_{n}\left(k\right)}\right).$$

The adequate situation where the frequencies $f_{i}$ of the generated sines $\Psi_{i}$ are multiples of $f_{0}$ leads to the following observation: To each *k*-coefficient is associated a frequency $f_{k}$, which is linked to the DFT computation frequency $f_{0}$ by $f_{k}\;=\;kf_{0}$, for k=0,…,P−1, and the following difference, which defines the resolution in the Fourier space:(18)$$\Delta f_{k}\;=\;f_{k+1}-f_{k}=f_{0}.$$

Choosing the generated frequencies as harmonics of $f_{0}$ leads to a match between $f_{i}$ and $f_{k}$ such that $f_{i}\;=\;f_{k}$, for i=k and i=0,…,P−1.

This observation is particularly interesting as with a given magnitude $M_{n}\left(k\right)$ is associated a frequency $f_{k}$, which corresponds to one and only one generated signal frequency $f_{i}$ on the electrode *n*. Therefore, the set of $\left[M_{n}\left(k\right)\right]$ for all *n* and *k* is one frame of the EIT data. The frame is acquired at the frequency $f_{0}$ and contains the full set of independent excitations and measurements to define the Dirichlet or the Neumann boundary conditions.

A first remark is that the highest frequency $f_{N}$ is constrained below the Nyquist limit: $f_{N}\leq f_{DAQ}/2$. A second remark concerns the phase shift, which describes the difference in radians when two or more alternating quantities reach their maximum or zero values. The phase shift of an electric signal passing through a material depends on the frequency [16]. In ONE-SHOT, the continuous generation of signals makes the phase shift independent of the Fourier magnitude. This is true because of the choice of the generated signals that are harmonics of the Fourier computation time window.

In standard EIT, the measurement associated with one excitation configuration corresponds to a voltage or current input over one or several periods of an alternative excitation at a given fixed frequency. The corresponding data are a set of tens of points per measurement, which multiplied by *M* (Equation (Equation 6)) gives the total number of samples per frame. On the other hand, in the Fourier space, the corresponding data for the same measurement becomes a single element $M_{n}\left(k\right)$ with *n* and *k* defined. Building a data frame with the Fourier elements reduces significantly the data size.

Finally, the determination of the frequencies for the application of a 16-electrode ONE-SHOT excitation strategy strongly depends on the acquisition rate $f_{DAQ}$ of the DAQ system and the choice for the number of measurement points *P* used to compute the DFT.

## 4. The Data Acquisition System

The concretization of an experiment to implement ONE-SHOT is based on the proposed hardware system presented in Figure 3. It is composed of three elements: the EIT sensor that contains the electrodes, the Printed Circuit Board (PCB) to distribute the signals, and the DAQ controller to manage the excitations and measurements.

Section 4.1 contains a discussion on the requirements for each component that led to the choice of this particular system. Then, in Section 4.2, the excitation and measurement strategy is applied to this system.

### 4.1. Selection of the Hardware

The starting point in creating a physical experiment for simultaneous EIT excitations and measurements is to establish the requirements on the DAQ system. The following details the issues and solutions found for the conception of its different components.

#### 4.1.1. The EIT Sensor

The EIT sensor is composed of a ring of electrodes in contact with the fluid and is non-intrusive. The electrodes must be insulated from each other and made in a conductive and robust material. In addition, the excitation and measurement signals at the electrodes are possibly of low amplitude.

The EIT sensor prototype used in our experiment is shown in Figure 1a. It was a 336 mm-long cylinder PMMA prototype with an internal diameter of 80 mm containing a set of 32 electrodes. The electrodes were chosen to be made of stainless steel. They were 150 mm long and 6 mm wide, and their surface was tangential to the inner pipe surface and collinear to the axis of the cylinder. In the experimental setting considered, only one over two, 16 electrodes were connected. The characteristics of the detector were chosen for future tests on static and dynamic flow measurements at the Laboratory of analytical Thermohydraulics and Hydromechanics of Core and Circuits (LTHC). More details on the design of the EIT for flow measurement applications can be found in [25].

The signal was routed to the electrodes using shielded coaxial cables. The EIT sensor provided Sub Miniature version A (SMA) adaptors to ensure the connection of the electrodes with the inner wire of the coax and the insulation of the shielding.

#### 4.1.2. The Printed Circuit Board

The main motivation in using the ONE-SHOT strategy is its high data acquisition rate. The natural choice for the hardware is to provide a very high sampling rate for excitation and measurement. As in Section 2, EIT relies on imposing or measuring either a voltage or a current on the boundary. However, the sampling rate for generation and acquisition of current is usually much slower than for the voltage.

A one-layer PCB (Figure 3) was designed to manage the voltage excitations and, in parallel, the current measurements. The currents passing through the electrodes (Neumann boundary conditions) were reconstructed from voltage measurements over 16 sense resistors of 100 Ω in the 16-electrode circuits on the PCB. This allowed current measurement using an analog voltage sampling at the DAQ level for a high sampling rate. Moreover, the PCB provided coaxial connectors and ensured the signal shielding with a ground connection.

#### 4.1.3. The DAQ Controller

The DAQ controller was required to perform 16 voltage excitations and 16 voltage measurements in parallel at high rate with great accuracy. The output configuration contained the generation of 120 sines at 120 different frequencies and a distribution onto the 16 electrodes under Equation (Equation 11). The input configuration computed the DFT online for each electrode signal at a high rate, based on the Fast Fourier Transform (FFT) algorithm. Finally, the DAQ controller was required to perform fast data storage.

Apart from the performance requirements, the DAQ controller must be robust and light weight for transportation purposes in future prospects of experiments on several hydraulic loops. It also has to be controllable at a distance for an application to high pressure and high temperature flow experiments.

The strict requirements of fast computations in both output and input for multiple channel operations, including FFT, suggest the use of a Field Programmable Gate Array (FPGA). In recent years, the number of logic blocks contained in an FPGA chip increased, and the generation of a large number of arbitrary signals became feasible.

A suitable device for the DAQ controller in the ONE-SHOT hardware system is the National Instruments CRIO-9039 (NI CRIO-9039: http://www.ni.com/pdf/manuals/375697d_02.pdf) controller, which includes a 1.91-GHz Quad-Core CPU, 2 GB of DRAM, and the Xilinx’s FPGA Kintex-7 325T. The FPGA includes 326,080 logic cells, 840 digital signal processing slices, and 16,020 block RAM elements. The main advantage of the FPGA is that it includes a large number of logic cells and large block RAM, which are required in our experiment to perform the operations described in Section 3. The adequate use of the FPGA allows fast DFT for a minimal data frame size, which allows efficient data transfer and reduces storage.

The CRIO-9039 DAQ controller includes a Linux host computer that operates ONE-SHOT with LabView 2017 Real-Time and FPGA. The monitoring of ONE-SHOT results in a large amount of data over numerous channels. The fast data monitoring is based on Direct Memory Access (DMA), a buffer to send the data to the host computer. In addition, the CRIO-9039 provides 8 slots in which Analog Output (AO) and Analog Input (AI) modules are connected.

#### 4.1.4. Output and Input Configuration

Regarding the necessity for fast and accurate operations, the AO was managed with the NI-9262 (NI-9262: http://www.ni.com/pdf/manuals/377139a_02.pdf) module to provide the voltage outputs. The six-channel module had a typical output voltage range of ±10.742 V, including an internal noise of 150 µV RMS per channel. In the system, three NI-9262 modules were connected to the 16-output excitations.

Concerning the AI, the NI-9223 (NI-9223: http://www.ni.com/pdf/manuals/374223a_02.pdf) module is suitable to provide voltage inputs. The NI-9223 includes four channels with a typical input voltage range of ±10.6 V with a noise of 229 µV RMS per channel. A total of four NI-9223 modules were required to connect the 16 input measurements.

Finally, the digitalized data format for both AO and AI modules were (20,5) fixed points: 20 bits allocated to the number including 5 precision digits. The data were operated at a sampling rate of 1 Mega Sample per second (MS/s) for both AO and AI, which is very competitive in terms of the performance requirements.

### 4.2. Practical Excitation and Measurement Strategy

In Figure 3, the excitation circuit is detailed in red. The host computer controls the FPGA to provide a fast generation of 120 sine signals. A second function inside the FPGA distributes the sines to the electrodes as in Equation (Equation 11). The 16 signals are transformed from 1 MS/s digital to analog signals using the NI-9262 modules and sent with coaxial cables to the electrodes through the resistors of the PCB.

In parallel, the measurement circuit, shown in green in Figure 3, contains AI modules to digitalize the voltages taken over the 16 resistances on the PCB. The FPGA-based DFT computation provides the signal magnitude and phase for each Fourier coefficient at the high rate of 1 MS/s. The electrode number *n* and Fourier coefficient *k* are associated with the magnitude value to provide its address: the host computer uses the address as *x* and *y* coordinates for the magnitude to build the data matrix. The FPGA sends the data with a Direct Memory Access (DMA) to ensure fast operations and reliability.

Considering the sampling rate of 1 MS/s of the DAQ system, one solution for the frequencies of the generated signals (Equation (Equation 11)) in the situation of a 16-electrode EIT sensor consists of $f_{i}=if_{0}$ for 1≤i≤N and N=120, as in Equation (Equation 7). Furthermore, the DFT computation can be chosen to be over P=512 points and results in a data frame acquisition rate of $1\times 10^{6}/512=1953$ fps. This choice implies the lowest sine frequency $f_{1}$ to be the same as the DFT computation frequency. The highest frequency $f_{N}=f_{120}$ is 234.375 kHz, below the Nyquist limit of 500 kHz for the considered system.

## 5. Analysis of the Error Propagation in DFT

Besides the hardware configuration, the measurement performances strongly depend on the DFT algorithm, which may have an influence on the uncertainty of the output data [26]. Consequently, there is a great interest in evaluating the systematic uncertainty in the DFT reconstruction. Two simulations are proposed, firstly in Section 5.1 without noise to assess the systematic uncertainty due to the discretisation of the DFT into P=512 points and, secondly, in Section 5.2, another simulation with a noisy signal to assess the propagation of noise in the DFT by quantifying the magnitude change in the Fourier space.

### 5.1. Noise-Free Sine Simulation

In the first step, the experiment focused on a limited number of frequencies. Let us impose 16 voltage excitation signals over the electrodes: 15 sines $V_{n}$ of frequencies $f_{i}$ and amplitude *A* such that:(19)$$V_{n}\;=\;A\sin\left(2\pi f_{i}p\right),\;\mathrm{for}\;n\;=\;1,\ldots ,15\;\mathrm{and}\;i\;=\;n$$
over the electrodes $E_{n}$. In addition, the definition of the ground (Equation (Equation 5)) suggests a drain voltage:(20)$$V_{Drain}\;=\;-A\sum_{i}\sin\left(2\pi f_{i}p\right)$$
imposed on the remaining electrode $E_{16}$ (Figure 4).

The lowest frequency is defined such that it corresponds to the frequency of the DFT computation time window (Section 3.4). The DAQ system described in Section 4 has a sampling frequency of 1 MS/s, resulting in $f_{1}$ = (512 *μ*s)^−1^ = 1953.125 Hz, which is also the resolution in the frequency space. Furthermore, the frequencies $f_{i}$ are the harmonics of $f_{1}$ such that $f_{i}\;=\;if_{1}$. The amplitude *A* is 0.8 V, so the drain signal remains in the NI-9262 AO module range limit of ±10.742 V.

The reconstruction of the magnitudes is shown in Figure 5. The systematic uncertainty due to the 512 points discretisation of the DFT was of the order of $O(1\times 10^{-15})$ V.

### 5.2. Noisy Sines Simulation

The second objective was to assess the magnitude of the distortion of the $k^{\mathrm{th}}$ harmonic due to the noise in the measured signal. In the time domain, the noisy signal ${\tilde{V}}_{n}\left(p\right)$ in the circuit of electrode $E_{n}$ contained a true signal $V_{n}$ and a Gaussian additive noise δ such that:(21)$${\tilde{V}}_{n}\left(p\right)\;=\;V_{n}\left(p\right)+\delta \left(p\right).$$

Since the Fourier transform is linear, the addition of noisy signal gives the following magnitude in the $k^{\mathrm{th}}$ harmonic of the measurement on the $n^{\mathrm{th}}$ electrode, as in Equations (Equation 13)–(Equation 16):(22)$${\tilde{M}}_{n}\left(k\right)={\left[{\left(\frac{1}{P}\sum_{p=0}^{P-1}\left(V_{n}\left(p\right)+\delta \left(p\right)\right)\cos\left(k\beta_{p}\right)\right)}^{2}+{\left(\frac{1}{P}\sum_{p=0}^{P-1}\left(V_{n}\left(p\right)+\delta \left(p\right)\right)\sin\left(k\beta_{p}\right)\right)}^{2}\right]}^{1/2}.$$

A simulation was proposed to assess the effects of a noisy signal by quantifying the magnitude change $\Delta M_{n}\left(k\right)\;=\;{\tilde{M}}_{n}\left(k\right)-M_{n}\left(k\right)$. The probability function, δ, of the Gaussian-distributed Gaussian noise pattern reads:(23)$$\delta \left(p\right)=\frac{1}{s\sqrt{2\pi }}e^{-\frac{1}{2}{\left(\frac{p}{s}\right)}^{2}}.$$

Several noise amplitudes with various standard deviation *s* were generated and added to the signal $V_{n}$ of Figure 4. The DFT coefficients of four different frequencies of low and high harmonic range are shown in Figure 6. On the abscissa, the Root Mean Square (RMS) of the generated noise $\delta_{RMS}$ is compared with the amplitude *A* of the generated sine as in Equation (Equation 8).

Due to signal filtering in the DFT computation, the error of the DFT magnitude for a given coefficient remains small by comparison with the original signal. However, the above simulation is based on white noise, which is well filtered by the DFT computation. Nevertheless, in the experimental case, it is important to detect possible constructing interference patterns in the noise, which could correspond to a generated frequency. The lengths of the cables in the DAQ system for instance have to be taken into account to not generate electromagnetic noise at one of the generated frequencies.

## 6. Preliminary Experimental Results with 15 Different Frequencies

As discussed in the Introduction, the imaging rate of the well-established X-ray tomography suddenly increased when considering several pairs of emitters and receptors for simultaneous measurements. The novel idea, which consists of applying simultaneous excitations and measurements in EIT, brings new challenges due to the fundamental differences existing between such hard field and soft field systems. Apart from the radical difference in solving the inverse problem (Section 2), multifrequency excitations and measurements have to be considered with the association of TDM in the ONE-SHOT strategy.

The discriminability of the raw data is discussed in Section 6.1 in two harmonic ranges. In Section 6.2, the noise measurement is shown with a discussion of the SNR of the measured data. Section 6.3 introduces the images reconstructed from a set of measurement data.

### 6.1. Raw Data

The ONE-SHOT excitation strategy has as its goal one single excitation for all independent pairs of electrodes, resulting in the generation of 240 positive and negatives sines (Equation (Equation 11)) of 120 different frequencies (Equation (Equation 7)) for 16 electrodes. Implementing ONE-SHOT is a challenging task as it requires fast voltage excitations of arbitrary signals and FFT computation of current measurements, all in parallel over numerous channels. Nevertheless, the simulation results in Section 5 are a great incentive to build the experiment as described in Section 4.

The preliminary results in this experiment are presented to prove the feasibility of using TDM in EIT, based on the simultaneous excitation of 30 positive and negative sines at 15 frequencies and the experimental DFT reconstruction of the same voltage excitation pattern as in Figure 4. Two frequency domains are investigated: the low harmonic range where the 15 positives sines are the 15 first harmonics of the DFT frequency $f_{0}$ (Section 3.4). Furthermore, the high harmonic range contains the harmonics 106–120: as discussed above, these higher frequencies are planned to be generated in the future development of a full simultaneous excitation set.

#### 6.1.1. Experimental Results in the Low Harmonic Range

An experiment was set up to measure a homogenous field of conductivity 300 µS/cm. The results in the low harmonic range (Figure 7) are the DFT magnitudes $M_{n}\left(k\right)/R$ of the 16 experimental measurement signals computed from P=512 data points, at a frequency of 1953.125 Hz for a data sampling of 1 MS/s. The results are shown as the current by including R=100
Ω, the resistance in the measurement channel on the PCB. The authors observed a good accordance with the simulations of Section 5. The comparison of the experimental results (Figure 7) with the simulated DFT reconstruction of pure sines (Figure 4) led to the following observations:
The readers familiar with EIT may expect the following phenomenon. In the experimental data, at the bottom of Figure 7 and Figure 8, non zero currents were measured over electrodes that were not excited at the corresponding frequency. Let us consider an electrode $E_{n}$ with 1≤n≤15 excited by a single sine of frequency $f_{i}$. In the Fourier space (Figure 7 and Figure 8), the measured current passing through an electrode gives non-zero values for other frequencies. The reason comes from the electric potential difference that appears between the excited electrode $E_{n}$ with an imposed potential $V_{n}$ with $-A\leq V_{n}\leq +A$ and a non-excited electrode $E_{m}$ whose potential is $V_{m}\;=\;0$ for a given frequency $f_{i}$ in the Fourier space (Figure 5). This effect is particularly large for the current in the adjacent measurement of the drain electrode in $E_{1}$ and $E_{15}$. These data depend on the electrical conductivity distribution inside the body and have to be considered as well.The magnitudes of the Fourier coefficients in the experimental results were lower for the mid-range harmonics than $f_{1}$ and $f_{15}$: Considering the finite electrical conductivity inside the system, the current resulting from the potential imposed between two neighboring electrodes was larger than the current resulting from the same potential imposed between two opposite electrodes in the circular shape of the EIT sensor [16].


#### 6.1.2. Experimental Results in the High Harmonic Range

The main prospect of the ONE-SHOT excitation strategy is to consider simultaneous excitations at 120 different frequencies for the EIT sensor of 16 electrodes. As in Section 3.4, the frequencies were chosen to be harmonics of the fundamental frequency $f_{0}$. The 120th harmonic $f_{120}$ was then 234.357 kHz. We investigated the performance of ONE-SHOT in this higher harmonic range.

The experimental measurement resulted in the high harmonic range from $f_{106}$–$f_{120}$ is shown in Figure 8. The discrimination of the signals by frequency offered equivalent performances to the low harmonic case. However, an important remark is that the signal amplitudes in the high harmonic range were lower than previously due to the impedance of water, which depends on the excitation frequency. The result was a weaker response in amplitude in the high harmonic range for the same conductivity change in the system. Investigations are ongoing for calibrating the amplitude of the generated sines to have a similar response in every harmonic for a future generation of a full set of 120 excitations.

### 6.2. Noise Measurement

A set of voltage magnitudes $M_{n}^{0}\left(k\right)$, defined as in Equation (Equation 16), was experimentally measured without any excitation to identify the noise spectrum. The results in Figure 9 show localized peaks in the noise amplitude beyond the high harmonic frequency domain.

The SNR of a signal measured at a given electrode *n* and tagged with a given frequency *k* is defined as follows:(24)$$SNR_{n}\left(k\right)=log_{10}\left[\frac{\sum_{\lambda }{\left[M_{n}\left(k\right)\right]}_{\lambda }/R}{\sum_{\lambda }{\left({\left[M_{n}\left(k\right)\right]}_{\lambda }/R\right)}^{2}+\sum_{\lambda }{\left({\left[M_{n}^{0}\left(k\right)\right]}_{\lambda }/R\right)}^{2}}\right]$$
where λ is the number of data taken for averaging. The noise in the low harmonic range resulted in an SNR of the raw signal of 69.1 dB in the lowest frequency magnitude in the excitation electrode channels when averaging the signal from λ=20 samples. In a non-excited channel, the signal was O(0.1) weaker, resulting in an SNR of about 60 dB depending on the signal magnitude. In the high harmonic range, the SNR was 59.6 dB for the highest harmonic.

The very high frequencies 250≤500 kHz, bounded above by the Nyquist limit, could be considered as excitation frequencies. However, a proper shielding of the DAQ system must be investigated. A large part of the noise was caused by electrical noise in the PCB. For this reason, a new PCB is being manufactured. Finally, the results showed that the noise completely covered the DFT systematic uncertainty (Section 5.1), which is completely negligible.

### 6.3. Image Reconstruction from 16 Datasets

The main objective of the present manuscript is the implementation of a method to obtain a very fast EIT data frame acquisition rate. From such data, the user can then reconstruct an image with any given reconstruction algorithm in post-processing, a step that we consider to be distinct from our main goal. As an example, we are interested in the image reconstruction of the above data. However, the datasets are incomplete by comparison with the ONE-SHOT excitation prospects of Equation (Equation 11). A solution consists of reconstructing the full dataset in the post-processing.

The results described above contained a complete set of excitations for the drain electrode. It is possible to consider another drain electrode and another frequency set to obtain another set of measurements, independent of location and frequency.

The rotation of the drain over the 16 electrodes resulted in 16×15=240 pairs of excitations. The symmetry suggested to consider only half of the data. The full dataset was deduced and contained the decomposition of 240 signals from 120 excitation pairs, measured over the 16 resistances of the PCB. Therefore, the full dataset included a total of 1920 elements. To sum up, the sequential excitations with the drain located on each of the 16 electrodes resulted in a dataset equivalent to the one expected by the full implementation of the ONE-SHOT method.

Several datasets were measured from the inclusion of non-conducting PMMA rods in the test section of Figure 1 filled with tap water with a conductivity of σ = 635 µS·m^−1^. The one-step least-squares iterative reconstruction method [27,28] was implemented, and the results are shown in Figure 10.

The results showed similar image reconstruction performances as another EIT system [16] based on TDM and using the same reconstruction algorithm. As usual in EIT, the sensitivity was much higher close to the electrodes. This effect resulted in sharp gradients when imaging the inclusions inserted at the edge.

Finally, the frame rate of the ONE-SHOT system can be changed by integrating the FFT over more or less data points. For instance, choosing the FFT to be computed over 512 points as considered in this article resulted in the frame rate of 1953 fps. In this case, the 120 frequencies can fill all the discrete values from 1953 Hz to the Nyquist limit of 500 kHz with Δf=1/512 µs = 1953 Hz the minimal gap between two neighboring values. The high frame rate was at the cost of a large frequency bandwidth.

## 7. Conclusions

This article assessed the feasibility of simultaneous excitations and measurements using a multi-frequency strategy for high-speed electrical impedance tomography, based on the ONE-SHOT excitation method. The motivations for increasing the data frame rate for EIT measurement followed by an overview of the mathematical aspect of EIT were presented in Section 1 and Section 2, respectively. The requirements for the hardware system based on the excitation strategy and measurement process, as well as the motivations to determine the number of electrodes were discussed in Section 3. An efficient hardware solution for implementing the simultaneous EIT excitation strategy was proposed in Section 4 with an analysis of the error and noise propagation through the measurement process in Section 5. Finally, experimental results were shown in Section 6 in the low and high harmonic ranges of the excitation signals.

Table 1 compares the performances of ONE-SHOT with a number of existing high-rate EIT systems. The description of the different hardware systems contains several characteristics that have to be considered while comparing the data frame rates, including the Number of Measurements per Seconds (NMS). Firstly, the AI sampling frequency directly impacted the data frame rate. Secondly, the data size was an important factor to maximize for a good conditioning of the inverse problem. Thirdly, the noise also directly impacted the image quality. The noise of the ONE-SHOT system was the value currently measured in a partially-shielded DAQ prototype. Future tests, especially including the new PCB, will result in better performances in terms of SNR.

The next experimental step is to measure a full scan by exciting all independent pairs of electrodes. For a 16-electrode EIT device, the number of frequencies becomes 120, bringing several new challenges. Mainly, the limit imposed by the Nyquist frequency of 500 kHz for the current system associated with the resolution in the frequency space from the choice of the DFT computation time window results in a maximum number of harmonics. Adding more frequencies may impose a longer time window, directly impacting the data frame acquisition rate. However, considering frequencies in the very high harmonic range may drastically affect the SNR due to higher resistivity and noise peaks. Furthermore, the limited physical space in the FPGA restricts the number of sine generators. The prospects of the performances of ONE-SHOT are also shown in Table 1. In these prospects, the DFT computed over measurements of P=256 points may increase the data acquisition rate, keeping $f_{120}=469$ kHz, below the Nyquist limit.

## Figures and Tables

**Figure 1 sensors-19-03679-f001:**
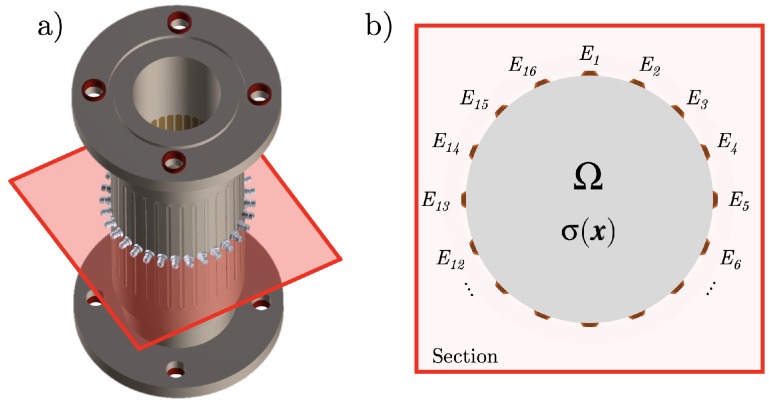
(**a**) Scheme of the EIT sensor prototype of height 336 mm, external diameter 120 mm, and inner diameter 80 mm containing 32 electrodes of length 150 mm and width 6 mm. In the current experiment, one over two, 16 electrodes are connected to coaxial cables with the SubMiniature Version A (SMA) connectors, also shown in the figure. (**b**) Representation of the 16 electrodes in the cross-section of the EIT detector.

**Figure 2 sensors-19-03679-f002:**
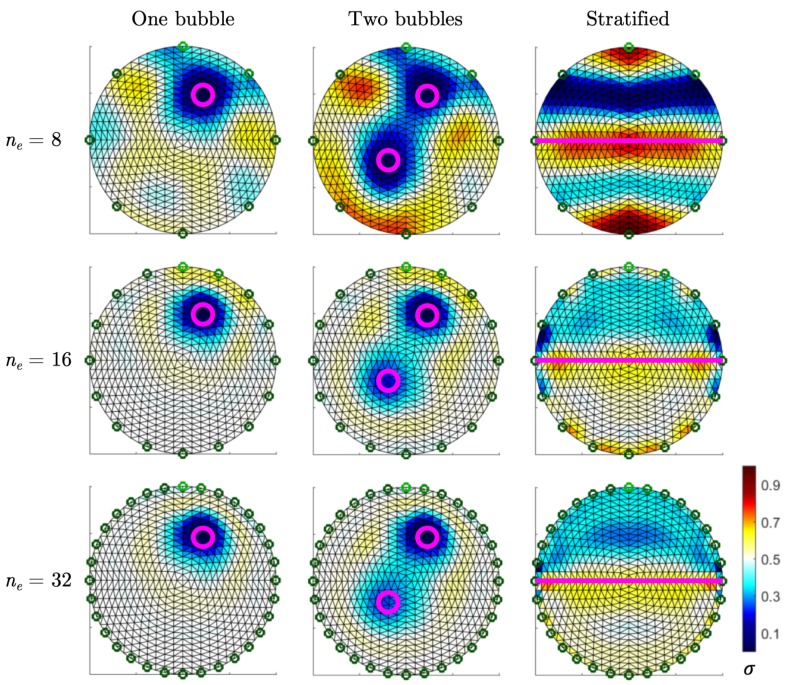
Image reconstruction using linear back projection from simulated data of bubbles in liquid for EIT detectors containing 8, 16, and 32 electrodes, represented with the green circles. On the left, one bubble of diameter 0.1 D, with D being the diameter of the pipe. In the middle, two bubbles of the same diameter. On the right, stratified flow. The gas–liquid interface is shown with the purple line.

**Figure 3 sensors-19-03679-f003:**
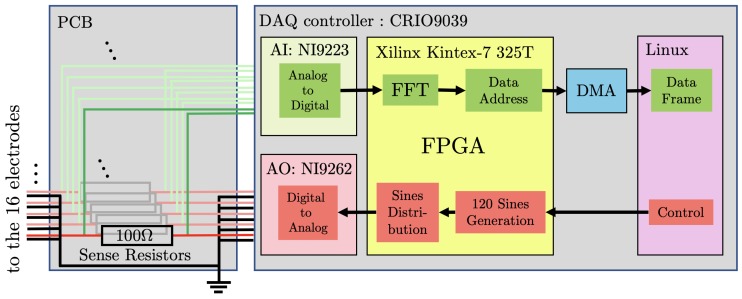
Layout of the DAQ system. On the left, details of the PCB that includes 16 independent circuits for voltage measurements over 100 Ω resistors. On the right, the FPGA-based CRIO-9039 configuration containing four NI-9223 AI modules, three NI-9262 AO modules, and a Direct Memory Access (DMA) buffer all controlled by the host, operated by Linux. In red, the excitation circuit. In green, the measurement circuit.

**Figure 4 sensors-19-03679-f004:**
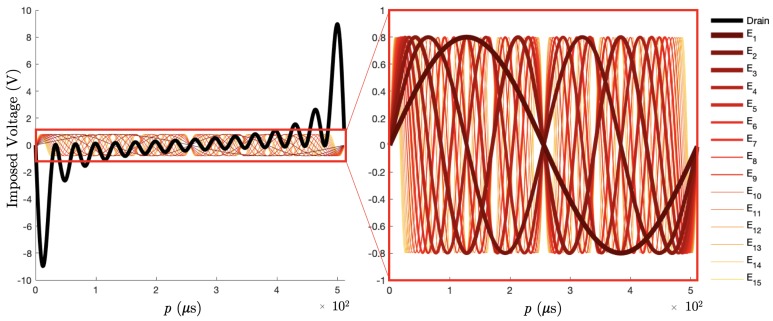
The 15 Excitation signals $V_{n}$ imposed on electrodes $E_{n}$ of frequencies $if_{1}$ for 1≤i≤15, and $f_{1}\;=\;{\left(512\;\mu s\right)}^{-1}$. The drain excitation signal is shown in black on the left plot.

**Figure 5 sensors-19-03679-f005:**
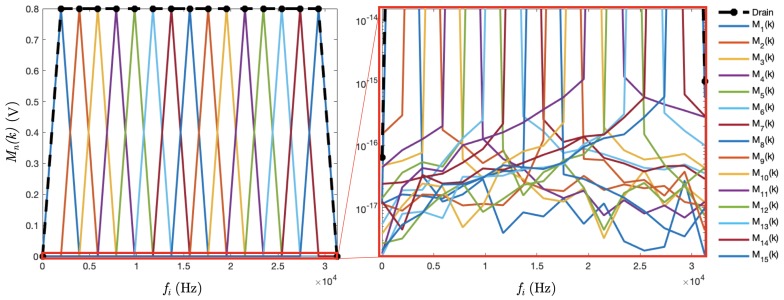
Simulation of the magnitudes of the 15 frequencies plus the drain at the 16 channels. On the right, rescaled zoom in the bottom part in the logarithmic scale of the DFT plot to show the tails of the peaks due to the discretisation onto 512 points.

**Figure 6 sensors-19-03679-f006:**
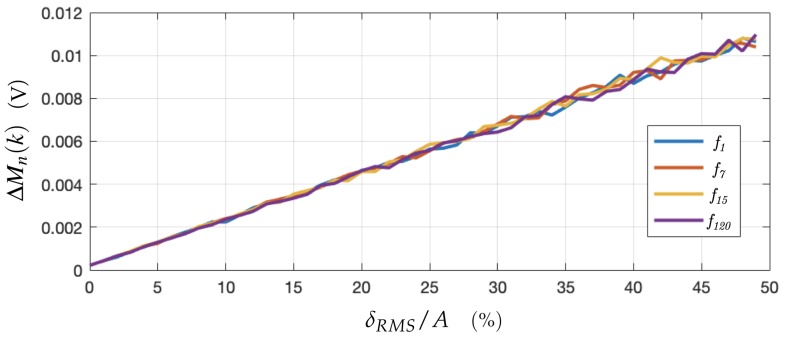
Propagation of the Gaussian white noise through DFT computation for four generated frequencies. The Noise RMS was computed from 1000 Gaussian white noises and averaged out. The four curves follow the same linear interpolation: $y\;=\;2.14\times 10^{-4}\;x+1.81\times 10^{-4}$.

**Figure 7 sensors-19-03679-f007:**
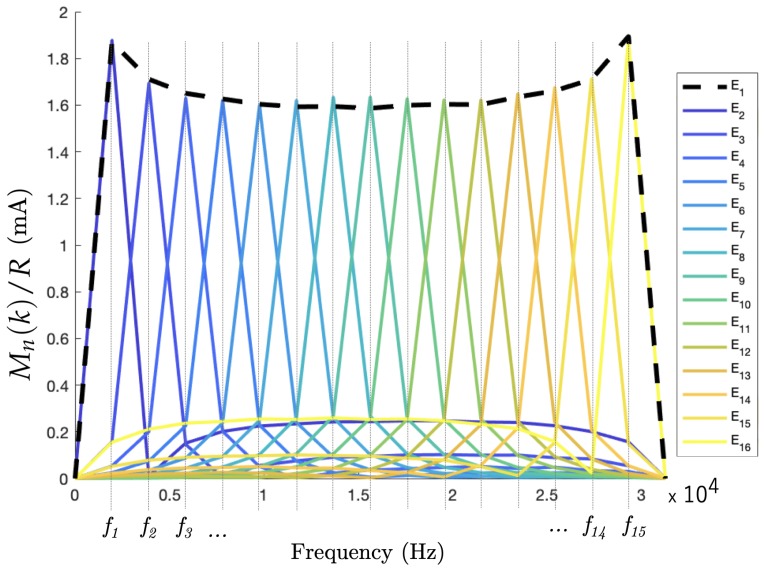
Measured current, represented as the magnitude of the DFT of 16 voltages over resistances in the excitation circuit of the 16 electrodes. This result corresponds to excitations in the low harmonic domain.

**Figure 8 sensors-19-03679-f008:**
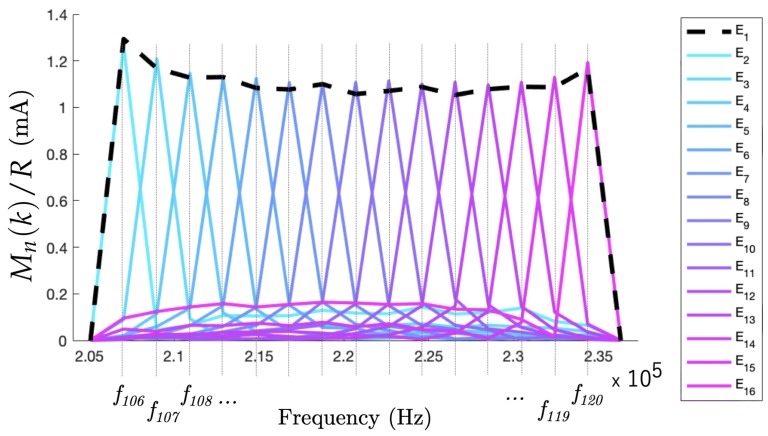
Magnitudes of the DFT from excitations in the high harmonic domain.

**Figure 9 sensors-19-03679-f009:**
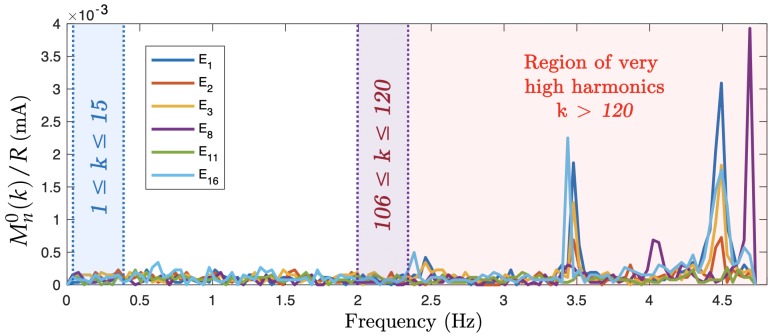
Noise measured over the full spectrum available for the sampling rate of 1 MS/s. The results show five of the 16 electrode channels. The low, high, and very high harmonic ranges are also shown.

**Figure 10 sensors-19-03679-f010:**
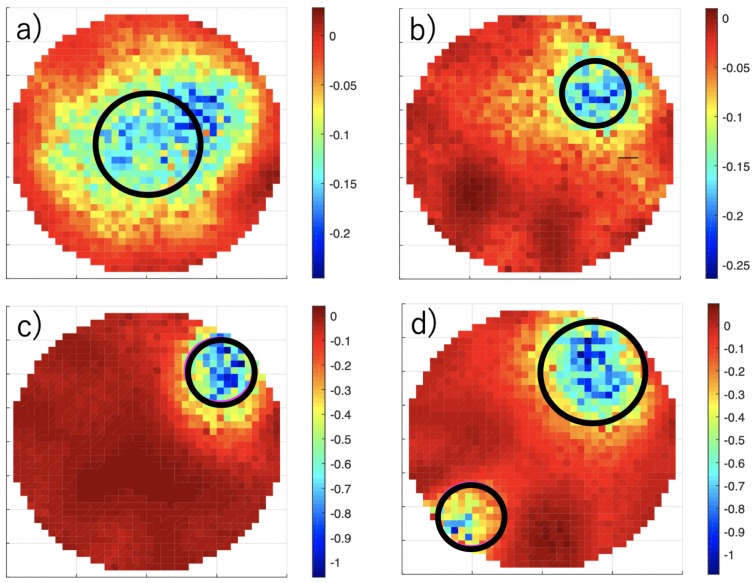
Electrical conductivity indicator function. Two non-conductive rods of diameter 20 and 30 mm, shown with black circles, are inserted in the test section filled with salt water. (**a**) A 30-mm rod at the center. (**b**) A 20-mm rod half way to the edge. (**c**) A 20-mm rod on the edge. (**d**) Two rods on the edge.

**Table 1 sensors-19-03679-t001:** Comparison of the performances of the ONE-SHOT simultaneous excitation difference with several high-rate EIT systems with single-frequency multiplexing strategies.

System	$f_{\mathrm{DAQ}}$ (MHz)	$n_{e}$	Data Size	NMS	Data Frame Acquisition Rate (fps)	Excitation Frequency (kHz)	SNR (dB)
Phantom [11], 2008	1	64	15 × 63	172,652	182.7	1–10,000	65.5–98.6
Phantom [12], 2015	10	16	16 × 16	28,160	110	97.65	90
ProME-T [16], 2017	2	16	16 × 120	1,599,360	833	15	60
ONE-SHOT, 2019							
(preliminary results)	1	16	16 × 15	468,744	1953.1	2–250	59.6–69.1
(prospects)	1	16	16 × 120	7,500,000	3906.3	4–469	-

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
