# Peer review of "On the Implementation of Simultaneous Multi-Frequency Excitations and Measurements for Electrical Impedance Tomography†"

_sensors, 2019, doi:10.3390/s19173679_

Round 1

Reviewer 1 Report

This paper is devoted to solve an important problem of Electrical Impedance Tomography (EIT) systems speed increasing for high-pressure and high-temperature fluid flows investigation. To solve this problem the innovative ONE-SHOT method developed by authors in previous works is considered, modeled and experimentally studied. The main feature of the ONE-SHOT method is a superposition of signals for simultaneous excitation in EIT system using a multi-frequency stimulation with discrimination of overlapping of signals using Frequency-Division Multiplexing techniques.

The introduction section of the paper proposes a wide enough review on recent techniques, algorithms and studies of high-speed multi-frequency EIT systems with 21 corresponding references. The fundamentals of ONE-SHOT method for high-pressure and high-temperature fluid flows investigation, proposed hardware system, analysis of the error propagation, simulation results and experimental data for noised signals are detail considered and discussed. Comparison of the proposed ONE-SHOT method with several high rate EIT systems with single-frequency multiplexing strategies is performed in the Conclusion.

The reviewer suggests that this paper will be very useful for specialists in the field of high-speed EIT systems. This paper can be accepted after following minor corrections.

1. In the line 159 M = 28 is incorrect for ne = 8. According to the expressions (8) and (9),
M = 224 for ne = 8.

2. According to the expressions (18) and (24), Mn(k) and DMn(k) are measured in Volts, not in Amperes as presented on the ordinate axis in Figures 5 and 6.

Author Response

 The reviewer suggests that this paper will be very useful for specialists in the field of high-speed EIT systems. This paper can be accepted after following minor corrections.

1.       In the line 159 M = 28 is incorrect for ne = 8. According to the expressions (8) and (9), M = 224 for ne = 8.

Thank you very much for your review and for the correction on this particular point.

2.       According to the expressions (18) and (24), Mn(k) and DMn(k) are measured in Volts, not in Amperes as presented on the ordinate axis in Figures 5 and 6.

The two figures are now corrected.

Reviewer 2 Report

This manuscript addresses a practical implementation of the ONE-SHOT method for multi-frequency excitation and measurement in EIT systems. The superposition of signals is discriminated using frequency-division multiplexing techniques. In my opinion, the following concerns should be addressed before further considerations.

- It is noticed that this work has been presented at the 9th World Congress on Industrial Process Tomography. The authors are suggested to clarify what is new with this article compared with the conference presentation. The novelties of this work should also be explained.

- I believe that imaging experiment is essential to demonstrate the performance of the designed EIT sensor system. Whereas the manuscript in its present form only presents image reconstruction simulations in the EIDORS environment when discussing “adequate number of electrodes” on Page 5. The experiment section only contains some simple current measurement results.

- In analyzing the error propagation, only Gaussian additive noise is considered, while it is well known that EIT system suffers from non-Gaussian non-additive noise and measurement errors. In this context, I believe that error propagation analysis using real collected data can be more compelling.

- Consistency of all frequency channels should be analyzed. Ideally, at different excitations, the boundary response voltage at the same relative position is consistent under the homogeneous object.

Author Response

- It is noticed that this work has been presented at the 9th World Congress on Industrial Process Tomography. The authors are suggested to clarify what is new with this article compared with the conference presentation. The novelties of this work should also be explained.

The context of the present article was indeed introduced at WICPT9. It concerns an experiment for measuring flow patterns with EIT in the hot-leg of a pressurized water reactor under accidental scenario. The conference article includes the very first steps of this experiment with the determination of the design of an EIT device to maximize the energy at the surface of the electrodes.

We realized after computing the flow evolution speed in such scenarios, that very fast EIT imaging is required to measure every changes. As a consequence, the present article includes the strategy that was developed afterwards and the presented results were mostly obtained after the conference.

Concerning the novelties of this work in the field of FDM EIT, the following paragraph has been added in the revised version of the manuscript l.76-88:

The ONE-SHOT approach brings several novelties in the field of FDM EIT. Firstly, in the excitation strategy of the previous systems, half of the electrodes are tagged with an excitation frequency and the other half is used for measurement: the electrodes are either used for excitation or measurement. The ONE-SHOT method provides current measurement in the excitation circuit, and each electrode is simultaneously used for both excitation and measurement. Secondly, in the previous systems the current excitation electrodes are tagged with a single frequency. On the other hand, ONE-SHOT introduces in this article an experiment where 15 excitation signals at 15 frequencies are imposed on a single electrode. It aims at demonstrating the applicability of FDM in this situation. Thirdly, a central argument in using FDM is the absence of transients at the electrode/electrolyte interface resulting in high data frame rate and low noise. The association of continuous excitations with a point-by-point synchronous Fourier transform is discussed to optimize the data acquisition speed. In addition, a hardware innovative methods leads to the implementation of every excitations and measurements, including real-time Fast Fourier Transform on a single FPGA chip.

 - I believe that imaging experiment is essential to demonstrate the performance of the designed EIT sensor system. Whereas the manuscript in its present form only presents image reconstruction simulations in the EIDORS environment when discussing “adequate number of electrodes” on Page 5. The experiment section only contains some simple current measurement results.

The main objective of the present manuscript is the implementation of a method to obtain a very fast EIT data frame acquisition rate. The user can then reconstruct an image with any given reconstruction algorithm from these data and this constitutes a postprocessing step that we consider to be distinct from our main objective. However, based on your comment and to illustrate this, an example is now provided in Section 6.3, which includes original data.

 - In analyzing the error propagation, only Gaussian additive noise is considered, while it is well known that EIT system suffers from non-Gaussian non-additive noise and measurement errors. In this context, I believe that error propagation analysis using real collected data can be more compelling.

In practical EIT, three causes to the noise are reported in [24]:

1         Measurement errors in the input and output voltages

2         Hardware problems

3         Modeling errors such as the approximation of the Neuman-to-Dirichlet infinite-dimensional operator to a discrete matrix in the electrode modeling of the inverse problem.

The article focuses on the EIT raw data obtained from FDM and hence, the first two contributions of the noise. The first contribution is discussed in section 4.1 and is originated from the hardware components of the DAQ controller. The second contribution is related to the noise appearing from the interaction between the electromagnetic field of the laboratory with the other hardware components such as the electrodes, the wires, the PCB or with the medium of the region of interest. Depending on their physical characteristics, the noise spectrum is not constant due to reinforced noise harmonics.

Our analysis is similar to the approach in the chapter 13 of [24]. In this reference, the simulation of noisy EIT raw data is based on the addition of a random matrix to the data matrix. The matrix elements are independent Gaussian random variables with zero mean and unit standard deviation. The standard deviation of the noise is adjusted by multiplying a constant to this matrix.

Based on your comment and the suggestion in the second part of your comment, we have added an analysis of the noise response from real measurements in the full spectrum, please see Section 6.2.

- Consistency of all frequency channels should be analyzed. Ideally, at different excitations, the boundary response voltage at the same relative position is consistent under the homogeneous object.

As in Section 6.1, the response of the system in measuring a homogenous conductivity field results in lower magnitudes for higher frequencies. This well-known result is related to the impedance dependency of water to the current frequency as given by the Bode analysis. This effect results in a lower magnitude difference between the homogenous and inhomogeneous measurements in the high harmonics range. We observe a lower sensitivity, giving smoother contrasts.

We measured the Bode diagram in the context of our experiment and used the relative impedance to calibrate the system response. The first attempt was to linearize the system response in post treatment but this method results in an amplification of the noise as well. Another calibration method is investigated, based on a non-constant excitation amplitude that depends on the frequency in a way to anticipate the system response. However, numerous developments and tests remain to be fully achieved before we can publish these results.

Reviewer 3 Report

The manuscript describes an interesting implementation for EIT but, in my opinion, it has a number of serious flaws that make it unpublishable at Sensors in its present form and contents.

- If the system is fully implemented, why not to report images acquired with it? Not to report images when describing a tomography system is quite unconvincing; particularly if superior performance is claimed. I guess the reality is that the system is not fully implemented. In this case the manuscript contents may be adequate for a conference publication but, in my opinion, it is not relevant enough for a journal publication.

- Both in the abstract and in the text, the reader is misled into believing that this is the first instance in which FDM is proposed for EIT.  To the best of my knowledge, such concept was originally proposed in the paper by Granot et al. (ref [19] in the manuscript). That paper is certainly cited in the manuscript but not in an appropriate way.  Proper recognition is needed and the differences (if any) between the “ONE-SHOT” concept and the concept proposed by Granot at al. should be explained.

- It is claimed a very superior performance in terms of frames per second but, what about frequency bandwidth? This important aspect is missing in the analysis. This is particularly relevant because excitatory frequencies as low as 2 kHz are proposed.

- Section 2 and parts of section 3 are known matter in the field of EIT and should be shortened. Or, at least, the contents should not be presented as if it was original matter develop by the authors.

- The definition of SNR must be clarified. For EIT systems it is possible to define SNR in different ways and, obviously, those different definitions yield different SNR values.

Author Response

 - If the system is fully implemented, why not to report images acquired with it? Not to report images when describing a tomography system is quite unconvincing; particularly if superior performance is claimed. I guess the reality is that the system is not fully implemented. In this case the manuscript contents may be adequate for a conference publication but, in my opinion, it is not relevant enough for a journal publication.

Thank you for reviewing the article and for your comments. We realized thanks to your feedback, that the original version of the manuscript does not fully support the presented results. The two main key points of the article are:

1         A theory to generalize the ONE-SHOT method: a simultaneous full set of excitations and measurement based on FDM for EIT.

2         A proof of feasibility by decomposing 15 signals simultaneously generated on a single electrode.

As mentioned in the conclusion, the objective of the experiment is to generate and measure simultaneously a total of 240 signals including 120 different frequencies. However, numerous problems remain before obtaining a fully implemented ONE-SHOT system, and operate fast EIT. Namely, extending the firmware used in this article from 15 frequencies to 120 frequencies does not fit inside a single FPGA chip. There are at least two solutions to this problem. Firstly, using several chips however the fast operation necessitates a highly performing synchronization strategy. Secondly, generating the signals point by point in a precomputed look-up-table, saved on the FPGA. The second option is considered. Another problem is that the full operation of ONE-SHOT at maximum speed necessitated high performance data buffer to handle up to 75 MB/s of data. This observation necessitates the implementation of a high performance data saving system.

In this context, the main objective of the present manuscript is the implementation of a method to obtain a very fast EIT data frame acquisition rate. The user can reconstruct an image using any given reconstruction algorithm from these data, in a post-processing step that step that we consider to be distinct from our main objective.

However, to provide an example, we have added images, reconstructed in post process from a collection of data obtained with the method described in the manuscript. For more detail, please refer to Section 6.3, which includes original data. To conclude, to our opinion, the manuscript reports an important milestone toward the objective of operating a fully implemented ONE-SHOT system.

 - Both in the abstract and in the text, the reader is misled into believing that this is the first instance in which FDM is proposed for EIT.  To the best of my knowledge, such concept was originally proposed in the paper by Granot et al. (ref [19] in the manuscript). That paper is certainly cited in the manuscript but not in an appropriate way.  Proper recognition is needed and the differences (if any) between the “ONE-SHOT” concept and the concept proposed by Granot at al. should be explained.

To the best of our knowledge, the concept of FDM in EIT applications was introduced in G. Teague “Mass flow measurement of multi-phase mixtures by means of tomographic techniques”, 2002. In the article of Granot et al., a similar excitation strategy based on FDM is considered. It includes a noise analysis, image reconstruction comparison with simulation, and confrontation of the results with time division multiplexing EIT and trigonometric excitation strategy.

The following paragraph is now reported in the introduction of the present manuscript (L76 to L88):

The ONE-SHOT approach brings several novelties in the field of FDM EIT. Firstly, in the excitation strategy of the previous systems, half of the electrodes are tagged with an excitation frequency and the other half is used for measurement: the electrodes are either used for excitation or measurement. The ONE-SHOT method provides current measurement in the excitation circuit, and each electrode is simultaneously used for both excitation and measurement. Secondly, in the previous systems the current excitation electrodes are tagged with a single frequency. On the other hand, ONE-SHOT introduces in this article an experiment where 15 excitation signals at 15 frequencies are imposed on a single electrode. It aims at demonstrating the applicability of FDM in this situation. Thirdly, a central argument in using FDM is the absence of transients at the electrode/electrolyte interface resulting in high data frame rate and low noise. The association of continuous excitations with a point-by-point synchronous Fourier transform is discussed to optimize the data acquisition speed. In addition, a hardware innovative methods leads to the implementation of every excitations and measurements, including real-time Fast Fourier Transform on a single FPGA chip.

 - It is claimed a very superior performance in terms of frames per second but, what about frequency bandwidth? This important aspect is missing in the analysis. This is particularly relevant because excitatory frequencies as low as 2 kHz are proposed.

The following paragraph has been added in Sec. 6.3 (L459 to L463):

Finally, integrating the FFT over more or less data points can modify the frame rate of the ONE-SHOT system. For instance, choosing the FFT to be computed over 512 points as considered in this article results in the frame rate of 1953 fps. In this case, the 120 frequencies can fill all the discrete values from 1953 Hz to the Nyquist limit 500 kHz with Df = 1/512 us = 1953 Hz the minimal gap between two neighboring values. The high frame rate is at the cost of a large frequency bandwidth.

 - Section 2 and parts of section 3 are known matter in the field of EIT and should be shortened. Or, at least, the contents should not be presented as if it was original matter develop by the authors.

The article has been modified based on your comment: The Sec. 2 content on the resolution of the inverse problem has been shorten and the comparison with X-ray tomography has been removed. In the Sec. 3, a sentence is added (L169) to underline what is new or not in the field of EIT.

 - The definition of SNR must be clarified. For EIT systems it is possible to define SNR in different ways and, obviously, those different definitions yield different SNR values.

Thank you, the SNR is now defined in eq. (24).

Round 2

Reviewer 2 Report

I believe that the authors have adequately addressed all my concerns. No further comments.